**www.cambridge.org/qrd**

# Hole migration in cytochrome P450

Harry Gray[1]  and Jay Winkler[2]

[1]California Institute of Technology Chemistry & Chemical Engineering Faculty, USA and [2]Beckman Institute, California Institute of Technology, USA

CYP102A1; CYP119; CYP158A2; CYP3A4; electron transfer

**Corresponding author:**
Harry Gray;
Email: hbgray@caltech.edu

## Abstract

The cytochrome P450 enzymes catalyze the hydroxylation of organic substrates by dioxygen. The high-potential reactive intermediate in cytochrome P450 catalysis, compound I (CI), has the capacity to deliver oxidizing equivalents (holes) to the side chains of tryptophan, tyrosine, and cysteine amino acids. Successful P450 catalysis requires that CI reacts more rapidly with a substrate than with these redox-active residues. The kinetics of hole transfer to tryptophan, tyrosine, and cysteine residues in four different P450 enzymes have been modeled using X-ray crystal structure coordinates and the semiclassical theory of electron transfer. Monte Carlo sampling of reaction driving forces has been used to account for uncertainties in the formal potentials of redox-active groups. The kinetics simulations suggest that the mean survival lifetimes of holes on the hemes range from ~100 ns to ~100 µs. Although hole transfer to the enzyme surface through redox-active amino acid reduces substrate oxidation efficiency, it can protect the enzyme from damage when reaction with substrate fails.

 **CAMBRIDGE**
UNIVERSITY PRESS

## Introduction

The cytochrome P450 enzymes (designated by the prefix CYP) comprise a superfamily of monooxygenases that catalyze the incorporation of one oxygen atom from dioxygen into an organic substrate with the second oxygen atom released as water (Ortiz de Montellano, 1985; Sono *et al.,* 1996). The enzyme active site is a heme axially ligated by thiolate from a cysteine (Cys) residue. Of the four equivalents required for the formal reduction of dioxygen, two arise from the organic substrate and two are delivered by co-substrate NAD(P)H.

In the generally accepted catalytic mechanism (Ortiz de Montellano, 1985; Sono *et al.,* 1996; Groves, 2003; Sevrioukova and Poulos, 2013), substrate binds in the distal heme pocket of the ferric enzyme, inducing axial water dissociation, spin change from low to high, and an increase in the $Fe^{III/II}$ formal potential (Sligar, 1976). Coupling reduction potential to substrate binding provides one means to regulate enzyme activity. Reduction of the substrate-bound ferric enzyme ($[Fe^{III}(S_{cys})P]$, P = protoporphyrin IX) by one equivalent from NAD(P)H produces a ferrous state that binds $O_2$, creating a ferric-superoxide species ($[Fe^{III}(O_2^-)(S_{cys})P]^-$). Delivery of a second electron and one proton generates a ferric hydroperoxide ($[Fe^{III}(OOH)(S_{cys})P]^-$) that undergoes O–O bond cleavage with proton capture to form compound I (CI) and a water molecule. CI, the key intermediate in catalysis, is a ferryl center coupled to a radical delocalized over the porphyrin and thiolate ($[Fe^{IV}(O)((S_{cys})P)^\bullet]$) (Green, 1999, 2009). CI abstracts an H-atom from substrate to produce compound II (CII, $[Fe^{IV}(OH)(S_{cys})P]$) wherein the porphyrin-thiolate radical has been reduced and the $Fe^{IV}$ oxo ligand has been protonated. Finally, the hydroxyl radical rebounds to the substrate, producing the oxygenated product and restoring the ferric resting state of the enzyme.

The CI intermediate naturally attracts considerable attention, owing to its capacity to abstract hydrogen atoms from saturated hydrocarbon substrates. Carbon–hydrogen bond dissociation energies fall in the 375–420 kJ mol$^{-1}$ range, requiring CI/CII formal potentials at pH 7 (E$^{o\prime}$) to be greater than 1.0 V *versus* NHE. Green estimates that E$^{o\prime}$(CI/CII) = 1.22 V at pH 7 in CYP158A2 from *Streptomyces coelicolor* (Mittra and Green, 2019); CI in this enzyme has the potential to abstract hydrogen atoms from substrates with C–H bond dissociation energies as great as 400 kJ mol$^{-1}$. The CYP158A2 CII/($[Fe^{III}(OH_2)(S_{cys})P]$) potential (pH 7) is reported to be 0.99 V *versus* NHE. The sidechains of at least three amino acids, tyrosine (Tyr-OH), tryptophan (Trp-NH), and cysteine (Cys-SH) fall in this range suggesting that, in the absence of reaction with substrate, CI and CII can participate in redox reactions with these amino acids. Tommos places E$^{o\prime}$(Tyr-O$^\bullet$/Tyr-OH) in the 0.92–0.99 V range (pH 7), and E$^{o\prime}$(Trp-N$^\bullet$/Trp-NH) = 1.03–1.09 V, depending on the degree of solvent exposure (Tommos, 2022). Glutathione (GSH), a cytoplasmic tripeptide of glutamic acid, cysteine, and glycine, has a thiyl radical potential of E$^{o\prime}$(GS$^\bullet$/GSH) = 0.94 V (Madej and Wardman, 2007), and serves as a model for E$^{o\prime}$(Cys-S$^\bullet$/Cys-SH). Clearly, the oxidizing equivalents (holes) stored in CI and CII are unstable with respect to transfer to surface-exposed Tyr-OH and Cys-SH residues.

The fidelity of coupling substrate oxidation to $O_2$ and NAD(P)H consumption varies widely among the P450 enzymes. The coupling efficiency in cytochrome P450$_{cam}$ (CYP101A1) with

camphor as a substrate is greater than 70% (Kim *et al.*, 2008) with some studies reporting 100% (Paulsen *et al.*, 1993). With nonnative substrates (*e.g.*, ethylbenzene), however, the coupling efficiency falls dramatically (5%), generating hydrogen peroxide as the major uncoupling product (Loida and Sligar, 1993). The most abundant human P450, CYP3A4, has extremely broad substrate specificity, owing to its role in xenobiotic metabolism (Guengerich, 2015). CYP3A4 catalyzed hydroxylations of testosterone, bromocriptine, and tamoxiphen have been shown to proceed with efficiencies of 10% or lower (Grinkova *et al.*, 2013). The three major P450 uncoupling pathways are: loss of $O_2^-$ from $[Fe^{III}(O_2^-)(S_{cys})P]^-$ (autoxidation); loss of $H_2O_2$ from $[Fe^{III}(OOH)(S_{cys})P]^-$ (peroxide shunt); and reduction of CI and CII to produce water from the second $O_2$ oxygen atom (oxidase shunt) (Gorsky *et al.*, 1984; Atkins and Sligar, 1988; Loida and Sligar, 1993; Grinkova *et al.*, 2013). The reducing equivalents required for the oxidase shunt pathway can be provided by Tyr-OH and Cys-SH residues, with Trp-NH serving as a conduit.

Decades of experimental investigations (Gray and Winkler, 2003, 2005; Winkler and Gray, 2014a) have demonstrated that thermodynamic driving force and electronic coupling between redox centers are the primary regulators of electron flow through proteins. This conclusion is fully consistent with semiclassical electron transfer (ET) theory (Marcus and Sutin, 1985; Marcus, 1993). Electronic coupling between redox centers depends on their separation distance and the composition of the intervening medium (Gray and Winkler, 2003). To preserve CI for substrate activation, one might expect a wide buffer zone around P450 hemes devoid of Trp-NH, Tyr-OH, and Cys-SH residues. Instead, we found that cytochromes P450 are replete with chains of Trp-NH and Tyr-OH residues extending from the heme to the enzyme surface (Gray and Winkler, 2015, 2016, 2021; Winkler and Gray, 2015), with the closest residue usually less than 10 Å away (Winkler, 2026).

Semiclassical ET theory provides a tool to estimate the survival times of CI and CII in structurally characterized P450s. Although good estimates are available (*vide supra*), formal potentials of CI, CII, and redox-active residues in a given enzyme are not known with precision. Consequently, we have used Monte Carlo sampling methods to predict CI survival times and hole flow pathways in four cytochromes P450: CYP158A2, CYP119, CYP102A1, and CYP3A4.

## Methods

Intraprotein ET rate constants ($k_{ET}$) were calculated according to semiclassical theory (Eq. 1) (Marcus and Sutin, 1985; Winkler and Gray, 2014b). The distance-decay parameter ($\beta$) was fixed at 1.1 Å$^{-1}$.

$$k_{ET}(\Delta G^o, \lambda, \beta, r) = 10^{13} e^{-\beta(r-r_o)} e^{\left\{-\frac{(\Delta G^o + \lambda)^2}{4\lambda k_B T}\right\}} \quad (1)$$

The absolute temperature ($T$) was chosen to match experimental conditions for a given protein and $k_B$ is the Boltzmann constant. Electron donor–acceptor distances ($r$) were taken from structure coordinates in the RCSB Protein Data Bank, and the contact distance ($r_o$) was taken to be 3 Å. The pairwise distances among redox-active heme, Trp, Tyr, and Cys residues were defined as the shortest separations involving: Fe or any nonhydrogen porphine atom of the heme; any ring atom of Trp heterocycles; any aromatic carbon atom or the OH group of Tyr; or the S-atom of Cys residues. Standard free-energy changes ($\Delta G^o$) were defined by differences in formal potentials of reaction partners. For Tyr and Cys residues to be considered redox-active, a proton acceptor (imidazole N-atom or

**Table 1.** Mean formal potentials used for redox-active residues in P450 enzymes

| | $E^o$ (V *vs.* NHE) | | | |
| --- | --- | --- | --- | --- |
| | Heme | Tyr | Trp | Cys |
| Buried | 1.0 | 1.0 | 1.1 | 1.05 |
| Buried and H-bonded to $H_2O$ | – | 0.95 | 1.05 | 1.0 |
| Surface exposed | – | 0.90 | 1.0 | 0.95 |
| | $\lambda_{11}$ (eV) | | | |
| | 1.5 | 1.0 | 0.8 | 1.0 |

carboxylate O-atom) had to be within 5 Å of the Tyr hydroxyl group or the Cys sulfur atom. All Tyr and Cys residues were considered redox active if the residue sidechain solvent accessibility was ≥20%. Solvent accessibility was determined using the Biovia Discovery Studio Visualizer using 240 grid points per atom with a probe radius of 1.40 Å. Mean formal potentials used for redox-active groups $\langle E^o \rangle$ are set out in Table 1. Each redox-active residue was assigned a set of $10^4$ potentials normally distributed about the mean potential with a standard deviation of $0.1\langle E^o \rangle$. The $E^o$ distributions were produced using the MATLAB (Mathworks, Inc.) normally distributed random number function (randn). Self-exchange ET reorganization energies ($\lambda_{11}$) used in the kinetics modeling are set out in Table 1. Reactions in which proton transfer accompanies electron transfer were assigned slightly elevated $\lambda_{11}$ values. Reorganization energies for cross reactions ($\lambda$) were taken to be the average of self-exchange reorganization energies for the two reactants (Marcus and Sutin, 1985).

Rate calculations were performed by solving numerically the set of coupled differential equations describing hole migration from heme to the thermodynamic product. Rate constants were calculated for ET between all pairs of redox-active residues according to Eq. (1). The reaction product was one or more oxidized redox-active residues, the composition determined by the set of potentials used. Calculations were performed for all redox-active residues at their mean potentials, and for the $10^4$ sets of normally distributed potentials (10,001 calculations in total). The heme hole lifetime ($\tau$) was defined as the time required for the heme hole population to drop to $1/e$ of its initial value. Additional details are provided in Supplementary Materials.

## Results and discussion

### CYP158A2

The genome of the actinomycete *S. coelicolor* A3(2) codes for 18 cytochromes P450 (Lamb *et al.*, 2002). CYP158A2 from *S. coelicolor* has been shown to catalyze oxidative C–C coupling of flaviolin, the first step in a polymerization process producing a pigment that has been suggested to protect microbes from UV radiation (Zhao *et al.*, 2005; Rudolf *et al.*, 2017). This enzyme was also used to estimate cytochrome P450 CI and CII formal potentials (Mittra and Green, 2019). We have used semiclassical ET theory and kinetics modeling to estimate the kinetics of hole migration from CI to oxidizable residues on the periphery of CYP158A2 (PDB ID 1S1F).

*S. coelicolor* CYP158A2 has 11 oxidizable residues (6 Trp, 4 Tyr, 1 nonligand Cys). Reaction of the enzyme with *meta*-chloroperbenzoic acid (*m*-CPBA) produces long-lived CII in high yield (Yosca *et al.*, 2013). The failure to observe CI in rapid mixing

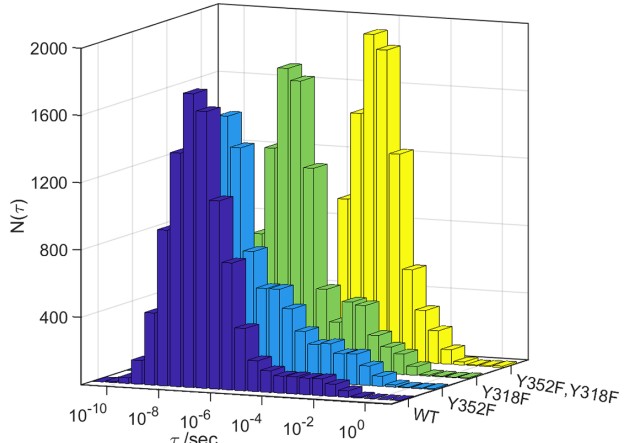

**Figure 1.** Distributions of calculated CI lifetimes (τ) for wild type and three mutant forms of CYP158A2.

experiments with *m*-CPBA was attributed to the rapid reduction of CI by Tyr352 and Tyr318 residues. Indeed, CI was observed in Tyr352Phe and Tyr352Phe/Tyr318Phe mutant enzymes (Yosca *et al.,* 2013) when mixed rapidly with *m*-CPBA at 4 °C. Our Monte Carlo kinetics modeling (see Supplementary Materials for details) places the mean CI 4 °C survival time in the wild-type enzyme at 170 ns with >75% probability of a survival time between 6 ns and 4.8 μs (Figure 1). We simulated the behavior of Tyr352Phe and Tyr318Phe mutants by removing one or both residues from the kinetics model. Removing Tyr352 from the model increases the mean CI decay time to 380 ns and substantially broadens the distribution on the slow decay side (Figure 1). The mean CI decay time for a Tyr318Phe mutant increases to 2.3 μs, and the mean decay time for the Tyr352Phe/Tyr318Phe double mutant is calculated to be 94 μs. In the rapid mixing experiments, the maximum CI concentration occurred at approximately 100 ms, consistent with a decay time on the order of 1 s. In the kinetics modeling of the double mutant, decay times greater than 100 ms

occurred in 61 (0.7%) of the calculations. The distributions of potentials for those 61 calculations did not show any substantial positive or negative bias away from the mean potential (Supplementary Materials).

To understand the trajectories of holes as they flow away from the heme, we considered the time dependence of hole populations on all redox-active residues as the system evolved from the initial state of one hole on the heme to the final state at thermodynamic equilibrium. We defined hole inflow as the maximum hole population found on a residue during the kinetics time course. We took the difference between the inflow magnitude and the value of the hole population at equilibrium as a measure of hole outflow. A plot of hole flow populations for CYP158A2 (Figure 2) illustrates that most holes from the heme flow through Tyr352 and Tyr318 before reaching equilibrium primarily on the four tyrosine residues. The central role of Tyr352 and Tyr318 is consistent with experimental observations that mutating these residues substantially increased CI survival (Yosca *et al.,* 2013).

### CYP119

CI from the cytochrome P450 from the acidothermophilic archaea *Sulfolobus solfataricus* (CYP119) (Park *et al.,* 2002) was first isolated and extensively characterized by Green's laboratory in 2010 (Rittle and Green, 2010). The CYP119 amino acid sequence contains: 14 Tyr residues, 12 of which we defined as redox active on the basis of the X-ray crystal structure (PDB ID 1IO7) (Park *et al.,* 2002); 5 Trp residues; and no nonligand Cys residues. Kinetics simulations with 10,001 different combinations of formal potentials for the heme and redox-active Tyr and Trp residues indicate a mean CI lifetime of 76 μs (Figure 3), comparable to the mean CI decay time calculated for the Tyr352Phe/Tyr318Phe double mutant in CYP158A2. The long CI lifetime in CYP119 is attributable to the 12.1-Å distance between heme and the closest oxidizable residue in CYP119 (Tyr46). This separation compares to 5.0 and 8.0 Å distances, respectively, between the heme and Tyr318 and Tyr352 in CYP158A2.

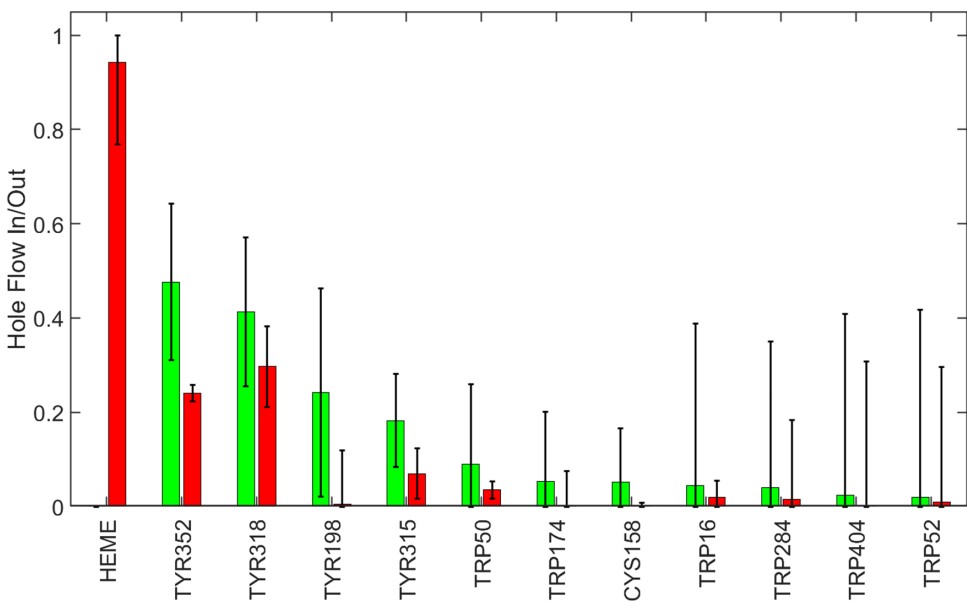

**Figure 2.** Calculated hole flow for redox active residues in CYP158A2. Green bars correspond to holes flowing into the residue and red bars correspond to holes flowing out. Error bars correspond to one standard deviation.

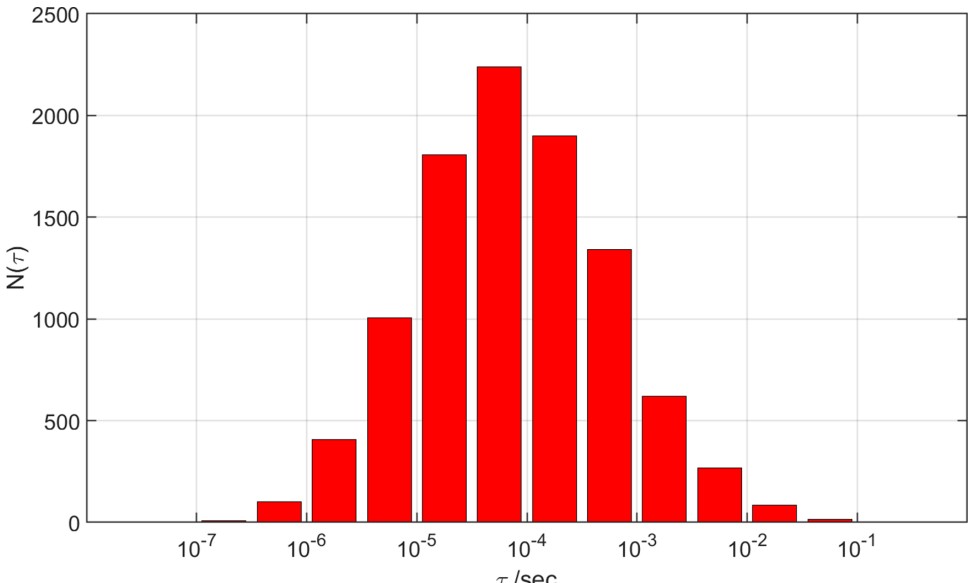

**Figure 3.** Distributions of calculated CI lifetimes ($\tau$) for wild-type CYP119.

The CYP119 CI intermediate decayed with a time constant of 100 ms (4 °C) in stopped-flow measurements with millimolar concentrations of enzyme and *m*-CPBA (Rittle and Green, 2010). Fewer than 0.05% of the calculated CI decay times are longer than 100 ms. The persistence of CI in CYP119 suggests that the CI formal potential is unusually low, or the potentials of redox-active Tyr and Trp residues are unusually high compared to nominal values set out in Table 1.

The plot of hole-flow statistics for CYP119 does not reveal any dominant residues participating in the hole migration process (Figure 4). The absence of a dominant hole-transfer pathway is consistent with the distribution of distances from the heme to redox-active residues. Nine redox-active residues have distances to the heme between 12.1 and 15.4 Å (Tyr46, Trp281, Tyr2, Tyr26, Tyr357, Tyr168, Tyr66, Tyr250, Trp147). The hole transfer rate constant to the most distant residue is expected to be just 37 times slower than that for transfer to the closest residue. This distribution of oxidizable residues creates many parallel paths for a hole to escape from the heme and migrate to the surface. Not surprisingly, CI decay in several simulated mutants (Tyr46Phe, Tyr357Phe, Tyr168Phe, Tyr46Phe/Tyr168Phe) produced no more than a two-fold increase in mean CI decay time. In this enzyme, then, it seems that CI lifetime is maximized by creating a buffer zone that is devoid of oxidizable residues around the heme.

## CYP102A1

The CYP102A1 enzyme from *Priestia megaterium* (formerly *Bacillus megaterium*) is a rare example of a self-sufficient P450 in which a reductase domain is fused to the heme domain so that delivery of

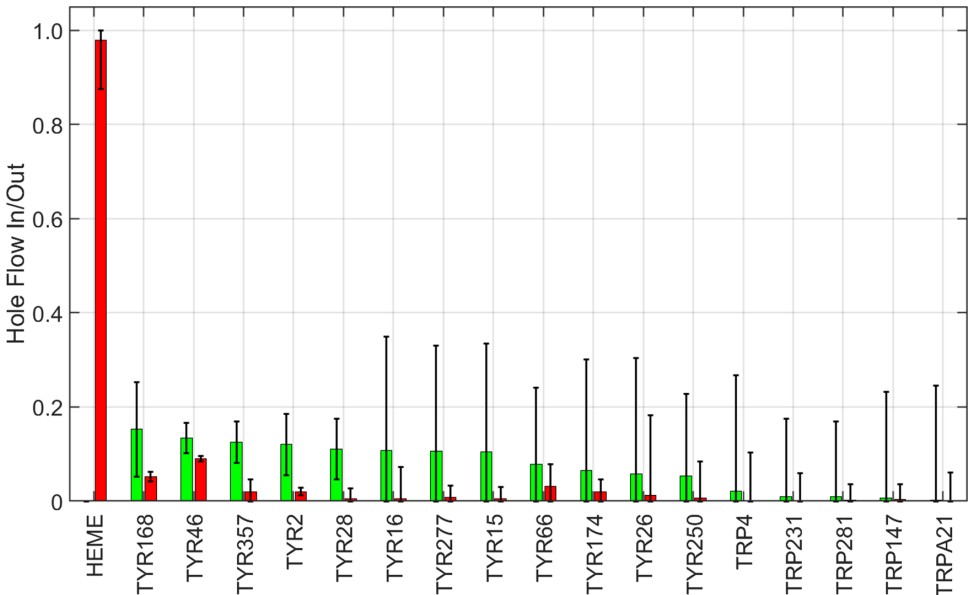

**Figure 4.** Calculated hole flow for redox active residues in CYP119. Green bars correspond to holes flowing into the residue, and red bars correspond to holes flowing out. Error bars correspond to one standard deviation.

electrons in the catalytic cycle is an intramolecular process (Whitehouse *et al.,* 2012). The enzyme was originally characterized as a fatty acid hydroxylase with a suspected biological function of detoxification of xenobiotic lipids produced by plants (Miura and Fulco, 1974; Thistlethwaite *et al.,* 2021). In prior work, we evaluated the activity of WT and mutant (Trp96His, Trp96His/Trp90Phe, Trp96His/Trp90Phe/Tyr334Phe, Trp96His/Tyr256Phe/Trp325Phe) CYP102A1 with *p*-nitrophenolate as substrate. Michaelis–Menten kinetics modeling revealed that enzyme activities defined by $k_{cat}/K_M$ remained relatively constant for wild-type and mutant proteins (Ravanfar *et al.,* 2023a). The coupling efficiency with this substrate was about 10% for wild-type and mutant enzymes, but very little of the uncoupling (<10%) produced hydrogen peroxide. The mutants, however, convert fewer substrate molecules before enzyme deactivation than the wild-type enzyme. These observations suggest that some or all of the mutated residues provide some degree of protection to the enzyme through the oxidase shunt pathway (Ravanfar *et al.* 2023a, 2023b, 2023b).

Using coordinates from the X-ray crystal structure of the CYP102A1 heme domain (PDB ID 2IJ2) (Girvan *et al.,* 2007), we evaluated the CI survival times for wild-type enzymes with 10,001 combinations of formal potentials for CI and 19 redox-active Tyr, Trp, and Cys residues. The mean CI survival time is 1.7 μs, consistent with the proximity of Trp96 to heme (7.3 Å). The flow of holes through these residues (Figure 5) implicates several that appear to play central roles in the migration of oxidizing equivalents to the enzyme surface. That the flow into Trp96 is virtually identical to the flow out indicates that holes transit through this residue on their route to surface residues. We performed kinetics

simulations for 8 simulated mutant enzymes and evaluated their mean CI survival times (Table 2). The critical role played by Trp96 is apparent in the ninefold increase in CI lifetime when this residue is removed. None of the other residues, however, has a comparable impact on CI survival, suggesting that after holes flow through Trp96, they can follow multiple routes to surface residues. This result is consistent with our observation that enzyme turnovers before deactivation are about the same for Trp96His, Trp96His/Trp90Phe, Trp96His/Trp90Phe/Tyr334Phe, and Trp96His/Tyr256Phe/Trp325Phe mutants.

### CYP3A4

The human enzyme CYP3A4 is one of the most abundant P450s expressed in the liver and small intestine and is believed to process at least half the drugs used clinically (Guengerich, 1999). Using coordinates from a high-resolution crystal structure of the N-terminally truncated enzyme (PDB ID 5VCC) (Sevrioukova, 2017), we simulated the hole migration kinetics in the wild-type enzyme. The closest redox-active residue to heme is Trp126 at a distance of 7.4 Å, followed by Tyr307 and Tyr432, both 10.1 Å away. The mean CI survival time is estimated to be 1.9 μs. Consideration of hole flow through redox-active residues implicates a pathway from the heme through Trp126, Cys98, arriving at surface-exposed Tyr99. The mean CI lifetime only increases by a factor of 2.4 if Trp126 is removed from the model. CI survival kinetics for several single and multiple mutants were simulated; the largest increase in lifetime was a factor of 2.6 for a Trp126His/Cys98Ser/Tyr99Phe mutant. As with CYP102A1, multiple pathways for hole migration from CI appear to be available in CYP3A4.

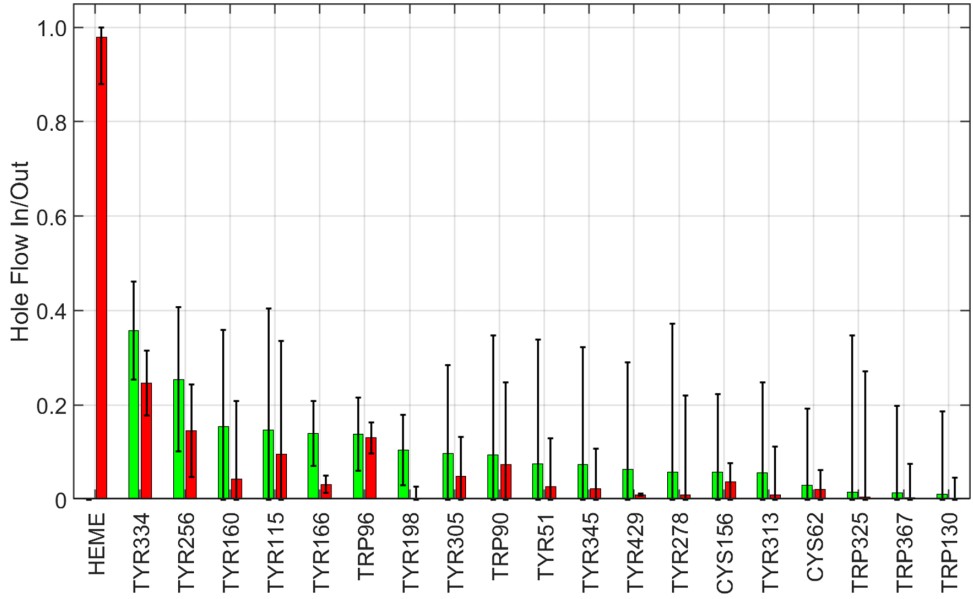

**Figure 5.** Calculated hole flow for redox-active residues in CYP102A1. Green bars correspond to holes flowing into the residue and red bars correspond to holes flowing out. Error bars correspond to one standard deviation.

**Table 2.** Mean CI survival times (⟨τ⟩, μs) calculated for wild-type and eight simulated CYP102A1 mutants

| WT | Trp96His | Trp90Phe | Tyr334Phe | Tyr256Phe | Trp325Phe | Trp96His Trp90Phe | Trp96His Trp90Phe Tyr334Phe | Trp96His Tyr256Phe Trp325Phe |
|----|----------|----------|-----------|-----------|-----------|-------------------|-----------------------------|------------------------------|
| 1.7 | 9.0 | 2.1 | 2.2 | 2.0 | 1.9 | 9.9 | 9.8 | 12.6 |

## Concluding remarks

Enzymes catalyze an astonishing array of complex chemical transformations. This fact is all the more remarkable given that these heteropolymers, constructed from a limited set of monomers and cofactors, can be extremely delicate and prone to inactivation. Of necessity, machinery is available to replace inactivated enzymes, but protein synthesis is a major energetic burden for cells (Lahtvee *et al.*, 2014). Mechanisms that limit or prevent enzyme deactivation might, therefore, be expected to provide a selective advantage to an organism.

Enzymes that catalyze reactions involving high-potential reactive intermediates would seem to be at particular risk of inactivation (Klinman, 2007). The cytochromes P450 are a case in point. Using reasonable parameters from electron transfer theory and enzyme structural coordinates, we estimated survival lifetimes for the key P450 reactive intermediate, CI. In the four wild-type proteins that we examined, this survival time ranged from ~100 ns to ~100 μs. For these enzymes to effect catalytic turnover, CI must abstract an H-atom from the substrate faster than it oxidizes a nearby redox-active residue. Whether the latter pathway represents the first step in enzyme deactivation or enzyme protection is an open question. The total turnover numbers in CYP102A1 decreased when redox-active residues near the heme were removed, which points to a protective mechanism (Ravanfar *et al.*, 2023a, 2023b). Demonstrating that these hole-migration pathways provide a selective advantage to the host organism is a far more challenging task.

**Open peer review.** To view the open peer review materials for this article, please visit http://doi.org/10.1017/qrd.2026.10021.

**Supplementary material.** The supplementary material for this article can be found at http://doi.org/10.1017/qrd.2026.10021.

**Data availability statement.** The authors will make all data and software available upon request.

**Author contribution.** Conceptualization: H.B.G., J.R.W.; Formal Analysis: J.R.W.; Funding Acquisition: H.B.G., J.R.W.; Methodology: J.R.W.; Software: J.R.W.; Writing: H B.G., J.R.W.

**Financial support.** Research reported in this publication was supported by the National Institute of General Medical Sciences of the National Institutes of Health under award number R01GM159231. Additional support was provided by the Arnold and Mabel Beckman Foundation.

**Competing interests.** The authors declare none.

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
