## [Reviewer Report]

The manuscript by Gray and Winkler contributes to understanding a fundamental challenge of the cytochrome P450 systems.

The P450s are one of the few enzyme systems that have the power to oxidize (e.g. hydroxylate) un-activated carbon positions. They do this through generation of an “active oxygen” at the heme center, formally an ‘oxene’ bound to iron(III), with a ground state ferryl porphyrin cation radical termed Compound I. With such a reactive species, how does the enzyme avoid self-destruction or oxidation of undesired substrates.

Many of the insights to answer this question were first realized by study of the bacterial Cytochrome P450cam (CYP101A1), that efficiently hydroxylated camphor as a first step in breaking the bicyclic monoterpene for carbon and energy utilization. Here the camphor displayed a high degree of complementarity to the enzyme active site, thereby displacing a water molecule as the sixth ligand, shifting the ferric spin state from S=1/2 to S=5/2. The coupling of spin state, binding free energy and change in redox potential was documented for CYP101A1 in 1976 by Sligar “Coupling of Spin, Substrate, and Redox Equilibria in Cytochrome P 450” Biochemistry 15, 5399-5406. Thus, in this case, the lack of substrate to bind and be metabolized, the enzyme could switch off the two electron transfers needed to convert atmospheric dioxygen to Compound I

But what happens when there is not such good pocket complementary to shut down generation of the hot Compound I oxidant? In particular, what about the hepatic P450s that have a very broad substrate specificity? Here the active sites are more accessible to solvent and the displacement of an iron-bound water may be incomplete at best. In these cases, substrate binding was still able to at least partially control the redox potential (Sligar, S.G., Cinti, D.L., Gibson, G.G., and Schenkman, J.B. (1979) “Spin State Control of the Hepatic Cytochrome P-450 Redox Potential.” Biochemical and Biophysical Research Communications 90, 925-932), and also the electron transfer rates into the heme (Backes, W.L., Sligar, S.G., and Schenkman, J.B. (1982) “Kinetics of Hepatic Cytochrome P 450 Reduction: Correlation with Spin State of the Ferric Heme.” Biochemistry 21, 1324 1330). A thermodynamic cycle explains that substrate must bind tighter to the reduced form of the enzyme to shift the potential to more positive values. It was shown that the absolute values of redox potential, however, did not correlate with binding free energy, but rather the ferric spin state equilibrium (Fisher, M., and Sligar, S. (1985) “Control of Heme Protein Redox Potential and Reduction Rate: Linear Free Energy Relation Between Potential and Ferric Spin State Equilibrium. ” Journal of the American Chemical Society 107, 5018-5019).

But another major difference between the bacterial P450 CYP101A1 and the hepatic enzymes was the efficiency of using the electrons transferred in from their redox partners. It was long known that there was a high degree of uncoupling, using the two electrons to reduce dioxygen to hydrogen peroxide, thus avoiding Compound I generation. This is another way of protecting the enzyme, although free peroxide can initiate Fenton chemistry and be destructive. Again, the first understanding came from the bacterial system

(Loida, P. J., and Sligar, S. G. (1993) “Molecular Recognition in Cytochrome P-450: Mechanism for the Control of Uncoupling Reactions.” Biochemistry 32, 11530-11538).

Yet another mechanism of uncoupling is the fast reduction of any Compound I generation at the P450 heme active site by two additional electrons to generate a second water molecule – a “cytochrome oxidase” activity. This was first shown by Coon and co-workers in the hepatic system (Gorsky, L. D., Kopp, D. R., & Coon, M. J. (1984) J. Biol. Chem. 259, 6812-6817) and extended in molecular interpretation in the bacterial system (Atkins, W.M., and Sligar, S.G. (1988) “Deuterium Isotope Effects in Norcamphor Metabolism by Cytochrome P-450cam: Kinetic Evidence for the Two Electron Reduction of the High-Valent Iron-Oxo Intermediate.” Biochemistry 27, 1610-1616).

So we have at least three mechanisms of dealing with a potentially hot oxidant: Control of electron input, branch off to produce hydrogen peroxide or reduce a hot Compound I to water. Initial thoughts were that maybe these differences were just between the “specific” bacterial systems and the promiscuous hepatic systems. But more recent work has shown that the human steroid metabolizing systems, even when in a nice membrane bilayer environment, can be highly uncoupled at both the peroxide and water branch points.

Grinkova, Y.V., Denisov, I.G., McLean, M.A., and Sligar, S.G. (2013) Oxidase Uncoupling in Heme Monooxygenases: Human Cytochrome P450 CYP3A4 in Nanodiscs.” Biochem. Biophys. Res. Comm. 430, 1223-1227).

What happens if these mechanisms do not operate? No redox switch. Insufficient proton access to generate peroxide or nucleophilic displacement of the heme ligand? Inefficient electron transfer to generate water from Compound I?

Over the past decade Winkler and Gray have documented a creative and important additional pathway. The Compound I oxidant could abstract from nearby amino acid residues and the resulting hole migration would deactivate the hot oxidative potential. Extensive previous work by these authors examined various cytochrome P450 structures to document common configurations that could provide a pathway for protection. In this communication the authors use semi-classical calculations to estimate the rates of hole migration. For the chosen examples, these range from 0.1 to 100 microseconds. The work is carefully executed and this quantitation is provides and important basis for understanding the overall metabolic mechanisms of the cytochrome P450s.

Further comments / input would strengthen the manuscript. How do the calculated hole hopping rates compare to the overall turnover number of the chosen native and mutant P450s (which vary greatly)? What about the degree of uncoupling for these enzymes, both peroxide release as well as the oxidase pathway. With estimates of the relative rates for productive and unproductive catalysis, is there a rationale for the contribution of this pathway to overall protection? Finally, what happens if none of these mechanisms work? What happens to the enzyme … aside from the case of BM3 and an engineered mutant that places an oxidizable residue near the heme (mimicking the heme oxygenases??) … is there evidence of suicide inactivation with a particular amino acid residue being modified (e.g. by mass spectrometry)?

There are a few specific editorial changes needed. First is Lines 5-7 in the introduction. There are no substrate electrons (maybe thinking of some dioxygenases?) == two electrons for normal P450 function. Four for the cytochrome oxidases and the uncoupling via reduction of Compound I.

The authors are free to use any of the references / texts in this review.